# Diabetes-Related Mortality in a Developing Country: An Exploration of Tertiary Hospital Data

**DOI:** 10.3390/jcm12206687

**Published:** 2023-10-23

**Authors:** Yanjmaa Sankhuu, Odgarig Altaisaikhan, Munkh-Od Battsogt, Oyuntugs Byambasukh, Altaisaikhan Khasag

**Affiliations:** 1Department of Endocrinology, School of Medicine, Mongolian National University of Medical Sciences, Ulaanbaatar 13270, Mongolia; s.yanjmaa@fchm.edu.mn (Y.S.); b.munkh-od@fchm.edu.mn (M.-O.B.); 2Department of Endocrinology and Diabetes, First Central Hospital of Mongolia, Ulaanbaatar 210648, Mongolia; 3Department of Health Research, Graduate School, Mongolian National University of Medical Sciences, Ulaanbaatar 14210, Mongolia; eph22e011@gt.mnums.edu.mn

**Keywords:** diabetes mellitus, mortality, cause of death, hospitalization, Mongolia

## Abstract

(1) Background: Given the growing global diabetes crisis, this study examined the causes of mortality in diabetic patients at a Mongolian tertiary care hospital. (2) Between 2017 and 2021, data from 100 individuals with diabetes (53% male, mean age 58.5 years, duration of diabetes, 9.6 years, HbA1c level, 9.7%, 11.1% type 1 diabetes) were reviewed. (3) Results: The predominant cause of mortality was sepsis, accounting for 65.0% of cases and emerging as a contributing factor in 75.0% of instances. Renal failure constituted the second leading cause of death, accounting for 19.0% of mortalities. Other contributing factors included chronic liver disease (6.0%) and ARDS (3.0%). Regarding sepsis, the individuals affected were relatively younger (57.5 ± 11.2 vs. 61.7 ± 11.2, *p* = 0.988), with a slightly higher prevalence among female patients (77.4%) and those with T1DM (81.8%), though these differences were not statistically significant (*p* > 0.05). Patients with sepsis exhibited lower BMI values (26.7 ± 4.1 vs. 28.5 ± 6.2, *p* = 0.014) and poorer glycemic control (9.8 ± 3.1 vs. 9.6 ± 5.1, *p* = 0.008); (4) Conclusions: This hospital-based data analysis in Mongolia highlights sepsis as the primary cause of mortality among diabetes patients in tertiary hospitals regardless of age, gender, or diabetes type while also indicating a potential association between a lower BMI, poor glycemic control, smoking, and the risk of sepsis.

## 1. Introduction

Diabetes is undeniably one of the most rapidly increasing diseases on a global scale [1]. According to the International Diabetes Federation (IDF), the latest statistics from the 10th IDF Diabetes Atlas (2021) reveal that 10.5% of the adult population aged 20 to 79 years currently has diabetes, with almost half of those affected unaware of their condition. Alarmingly, the IDF’s projections indicate that by the year 2045, approximately one in eight adults, totaling around 783 million people, will be living with diabetes, reflecting an alarming 46% increase [2,3]. As diabetes profoundly influences mortality, it becomes increasingly imperative to unravel the primary causes of death among individuals affected by this condition. Countless studies have delved into the intricate relationship between diabetes and mortality, scrutinizing both overall mortality rates and in-hospital occurrences [4,5,6,7,8]. For instance, Pouya et al. discovered that diabetes contributes to approximately 11.3% of global adult deaths, with variations across different regions [4]. In a similar vein, Jae Jeong Yang et al. illuminated the association between diabetes and a 1.89-fold amplified risk of all-cause mortality, with marked spikes in mortality due to diabetes itself, renal ailments, coronary heart disease, and ischemic stroke [5]. The evolving landscape of mortality rates among populations with diabetes, as highlighted by Lei et al., reveals divergent trends, with mortality rates receding in Europid populations while surging among Asian counterparts, thereby accentuating the complexities of diabetes in developing nations [6]. Building on the bedrock of prior research that underlined heightened in-hospital mortality among diabetic patients and identified culprits such as cerebrovascular maladies and infections as common causes of death [7,8], it is increasingly imperative to dissect the issue further.

As diabetes continues to pose a growing threat, even in developing countries such as Mongolia, where its prevalence has tripled in the last two decades, understanding the main causes of mortality in diabetic patients becomes paramount. The prevalence of diabetes increased from 3.2% in 1999 [9] to 8.3% in 2020 [10]. Cardiovascular disease (CVD) has been the leading cause of death since 1992. In 2019, 32.1% of all deaths in Mongolia were due to cardiovascular disease [11]. However, as country-based data are limited, our study focused on a hospital-based analysis. This approach is essential for identifying key factors and improving healthcare systems to address diabetes-related mortality challenges. Therefore, the primary objective of this study was to investigate the main causes of mortality among diabetic patients admitted to a tertiary care hospital in Mongolia from 2017 to 2021. The exclusion of data from the COVID-19 pandemic years, from 2021 to 2023, was a prudent decision to stave off potential biases. By setting the stage for a robust hospital registration system and conducting a comprehensive analysis, we endeavored to offer insights that can be wielded to transform healthcare strategies and diabetes management in developing countries, rendering them better equipped to tackle the main causes of mortality among diabetic patients.

## 2. Materials and Methods

### 2.1. Data Source and Study Population

In this study, a comprehensive dataset was collected from the First State Central Hospital Mongolia, a prominent tertiary healthcare institution established in 1965. Throughout its history, the hospital has played a crucial role, serving as the largest medical facility in Mongolia until the year 2000, and it continues to be one of the country’s leading hospitals, providing medical services to patients from across the nation. The hospital, operating at a third-tier level, specializes in managing complex cases referred from district or provincial hospitals, excluding trauma and burn cases, which are directed to specialized centers. In Mongolia, there are dedicated cancer and stroke centers. The hospital provides comprehensive care for a diverse range of diseases, offering services in both outpatient and inpatient settings. This includes managing conditions, such as conducting diabetic follow-ups, and addressing critical cases such as stroke or cancer. The hospital is well-equipped with specialized units and facilities, including a dedicated stroke unit and emergency rooms.

To assemble the necessary data for analysis, this study specifically concentrated on hospital discharge information for cases in which diabetes was a significant factor contributing to the patient’s condition. This study covered a substantial period, ranging from 1 January 2017 to 31 December 2020, to capture a comprehensive snapshot of diabetes-related mortality within the hospital. Ethical approval for the study was obtained from the Medical Ethical Committee of the Mongolian National University of Medical Sciences (METc2020/03), and the study was conducted in accordance with the Declaration of Helsinki.

### 2.2. Data Collection and Variables

Through systematic and detailed data categorizations, this study aimed to gain valuable insights into both the primary and secondary causes of mortality among diabetic patients admitted to the First State Central Hospital Mongolia over a four-year period. This comprehensive analysis sought to shed light on the critical aspects of diabetes-related mortality, ultimately contributing to the improvement of healthcare strategies and patient outcomes in managing diabetes in Mongolia and beyond.

The data were categorized Into two main variables: “Single Cause” and “Contributory Causes”. Under the “Single Cause” category, the primary cause of death was recorded for each patient. In cases in which individuals had multiple causes of death, they were classified under “Multiple Cause” cases within the “Single Cause” category. The research team conducted a thorough translation of the International Classification of Diseases, Tenth Revision (ICD10), codes into the specific causes listed in Table 1. Infections were predominantly referred to as sepsis arising from acute infections such as facial and body acne, diabetic foot alterations, and gastrointestinal inflammation, while gastrointestinal bleeding was excluded from this category. Additionally, the “Other Miscellaneous” category encompassed isolated or rare cases recorded, such as adrenal insufficiency and gastrointestinal blockage.

To further explore the nuances of diabetes-related mortality, the investigators developed a second variable named “Contributory Causes”. This variable involved a detailed breakdown of the multiple causes found within the “Single Cause” category. By segregating these multiple causes and identifying specific factors contributing to mortality, the study aimed to discern patterns and associations that might shed light on the complexities of diabetes-related deaths. For instance, if an infection presented as secondary cause of death along with other conditions, it was classified as one of multiple causes in the “Single Cause” category. Subsequently, the study team added these cases to the infection category within the “Contributory Cause” variable separately. This approach allowed for a more comprehensive understanding of the role infections play in diabetes-related mortality, providing valuable insights into their impact as both primary and contributory factors to patient outcomes.

After conducting a detailed analysis, sepsis was identified as the primary cause of mortality. To further understand the underlying factors contributing to sepsis, we categorized cases into different types, including respiratory (e.g., pneumonia, bronchitis, and influenza), abdominal (e.g., appendicitis, peritonitis, and diverticulitis), urinary (e.g., urinary tract infection, pyelonephritis, and renal abscess), and skin and soft tissue infections (e.g., cellulitis, abscesses, infected wounds, and paraproctitis), as well as cardiovascular (e.g., endocarditis, myocarditis, and pericarditis) and central nervous system infections (e.g., meningitis, encephalitis, and brain abscesses). It is important to note that due to the absence of bacterial, viral, and fungal infections, these were not considered in this analysis.

### 2.3. Statistical Analysis

The general characteristics of the study population were expressed as means with standard deviations (SDs) or means with standard errors of the mean (SEMs) based on the distribution of variables, and as numbers with percentages (%) for categorical variables. To compare the differences between groups, parametric and non-parametric tests were used. Parametric tests, such as Student’s *t*-test and an ANOVA, were applied to normally distributed continuous variables. For categorical variables, Pearson’s chi-squared test was employed.

All statistical analyses were performed using IBM SPSS V.28.0 (IBM, Chicago, IL, USA), and the level of statistical significance was set at *p* < 0.05 for all tests.

## 3. Results

During the research period (2017–2021), a total of 100 people were registered. Among these, 53.0% (n = 53) were male. The descriptive statistics for the variables are as follows: the mean age was 58.5 years ± 11.3, ranging from 32 to 84 years, and the duration of diabetes ranged from 0 to 37 years, with a mean of 9.6 years. The data analysis revealed that mortality could occur at any age and following any duration of diabetes, with no specific age group or duration of diabetes showing immunity to mortality risks (Figure 1).

However, among the study participants, the most prevalent age group was found to be individuals aged 51 to 67 years, covering the 25th to 75th percentile range. Similarly, the duration of diabetes was most commonly found to be between 2 and 13 years, also encompassing the 25th to 75th percentile range. The body mass index (BMI) values ranged from 17.3 to 43.8, with a mean of 27.1 ± 4.3. Glycated hemoglobin (HbA1c) data were available for 97 participants, showing HbA1c levels ranging from 4.6 to 26.0, with a mean of 9.7 ± 3.6%. In the treatment of diabetes, 15% of the patients relied on multiple daily doses of insulin, while 36% used oral glycemic drugs; the remaining individuals utilized a combination of both approaches. Among the prescribed medications, sulfonylureas were the most commonly used, followed by biguanides, incretin-based drugs, and sodium–glucose cotransporter-2 inhibitors (SGLT-2I, Table 2). Notably, in this population, the prevalences of smoking and hypertension were higher, accounting for 51.0% and 61.0%, respectively. The hospital stay durations varied from 0 to 103 days, with a mean duration of 12.5 days.

We conducted a comparative analysis between male and female patients, as outlined in Table 2. It was observed that female patients exhibited an advanced age, a higher BMI, and were less frequently smokers when contrasted with their male counterparts. Notably, these differences proved to be statistically significant, with *p*-values of 0.022 and 0.009, respectively. Other parameters, such as the duration of diabetes, the type of diabetes, glycated hemoglobin levels, diabetes treatment, hypertensive status, and length of hospital stay, showed no significant differences between the two groups.

The study delved into the various factors influencing mortality among patients with diabetes, shedding light on both primary (single) and contributing causes, as well as distinctions based on gender (Table 1). Notably, sepsis emerged as the predominant direct cause of death, contributing to a significant 65.0% of all participant fatalities. It is worth highlighting that these infections not only stood out as the primary direct cause but also constituted the most prevalent contributing factor to mortality, accounting for a substantial 75.0% of overall deaths. Moreover, when considering the prevalence of sepsis, it became evident that this condition affected women slightly more profoundly (77.4%) than men (72.3%). Another substantial contributing factor to mortality was chronic renal failure, which was responsible for 19.0% of all deaths. This condition exhibited a more pronounced impact on women (23.4%) compared to men (15.1%). In terms of specific contributory factors, acute respiratory distress syndrome (ARDS) played a role in 3.0% of deaths, with a slightly elevated incidence among women (3.8%) compared to men (2.1%). Chronic liver disease surfaced as a contributor to 6.0% of the total deaths, with a noticeable prevalence among women (8.5%) relative to men (3.8%). Malignancy was identified as a contributor in 4.0% of deaths, showcasing a higher occurrence in men (5.7%) as opposed to women (2.1%). Similarly, cerebral vascular accidents accounted for 6.0% of the fatalities, with a marginally increased frequency among women (8.5%) in comparison to men (3.8%).

In this study, 11 participants (11.0%) had T1DM, while the majority, 89.0%, had T2DM. Notably, the analysis revealed no significant disparities in terms of contributing factors to mortality causes. However, it is worth noting that sepsis demonstrated a slightly higher occurrence in individuals with T1DM compared to those with T2DM. Renal failure emerged as a common factor across both types of diabetes, implying that it contributed to mortality in both cases. Notably, diabetic ketoacidosis (DKA) was more prevalent in participants with T1DM. Additionally, the study noted that the participants with T2DM had higher incidences of various other miscellaneous factors contributing to mortality (Figure 2).

When examining the causes of mortality and their relationship to age and the duration of diabetes, patients with coronary artery disease had a longer diabetes duration of 17.0 years compared to those without coronary artery disease, who had an average diabetes duration of 9.4 years (*p* = 0.026). Additionally, patients with chronic renal failure were significantly older, with an average age of 66.3 years, compared to those without chronic renal failure, who had an average age of 58.4 years (*p* < 0.001).

Sepsis was the main cause following an analysis. There were significant differences in BMI and HbA1c levels between the groups. The sepsis patients exhibited lower BMIs and poorer glycemic control (higher HbA1c), hinting at a potential connection to the risk of sepsis (*p* < 0.05). Additionally, the sepsis group exhibited a higher frequency of smokers (*p* = 0.024). Age and diabetes duration showed no significant differences between the sepsis and non-sepsis groups. However, the sepsis patients tended to be younger, with shorter diabetes durations and a higher proportion of males. These differences were not statistically significant (Table 3). Though not significant, the sepsis patients had a slightly higher occurrence of T1DM. Notably, the sepsis patients had significantly shorter hospital stays compared to non-sepsis patients, implying different healthcare needs and outcomes for the two groups.

Upon further analysis, sepsis emerged as the leading cause of mortality. One-third of sepsis cases (29.5%) were attributed to skin and soft tissue infections, making it a significant contributor (Figure 3). Additionally, urinary tract infections (UTIs) and abdominal infections were prominent causes, accounting for 16.7% and 15.4%, respectively. Delving deeper into skin and soft tissue infections, cellulitis (52.2%), and paraproctitis (21.7%) were identified as primary culprits. In UTIs, pyelonephritis (53.9%) and renal abscesses (23.1%) were the predominant issues. In the realm of abdominal infections, cases included cholecystitis, pancreatitis, appendicitis, and others. Notably, infections related to malignancies were prevalent, constituting 10.3% of cases. Respiratory infections primarily manifested as pneumonia (71.4%). Interestingly, cardiovascular infections were not initially diagnosed as myocarditis or similar conditions but were associated with infarcts, possibly as a consequence. Similar scenarios were observed in central nervous system infections, which often occurred post stroke. Other contributing factors encompassed bone and joint infections.

## 4. Discussion

Diabetes has emerged as a pressing global health concern, with its prevalence on a steady incline, even in regions such as Mongolia, where it has tripled over the last two decades [9,10]. However, the absence of a comprehensive registration system in our country has left us devoid of specific in-hospital mortality data for diabetic patients. Discerning the primary causes of mortality among these patients is of paramount importance to enhancing their outcomes. With this goal in mind, our study sought to examine the mortality rates among diabetic patients admitted to a tertiary care hospital between the years 2017 and 2021.

Our findings underscore the pivotal role of infections in the mortality of the study participants, both as contributing factors and sole causes. In addition, our analysis reveals other noteworthy contributors to these fatalities. Notably, chronic renal failure, acute respiratory distress syndrome (ARDS), and chronic liver disease emerged as significant contributory causes. Furthermore, malignancy and cerebral vascular accidents (strokes) were identified as both contributory factors and primary causes. Interestingly, research conducted in tertiary medical facilities in India echoed similar patterns, highlighting infections and sepsis (40.9%), alongside chronic renal failure (33.6%), as the primary culprits behind the mortality of diabetic patients [12]. Correspondingly, a study conducted in Peru found renal failure (38.1%), infections (35.7%), and cerebral vascular accidents (16.7%) to be the major drivers of mortality [13]. Moreover, research from China indicated that end-stage renal disease mortality rates were twice as high among diabetes patients as in non-diabetic individuals [14]. In developing countries, sepsis and renal failure were the predominant causes of mortality among diabetic patients in hospital settings, while developed nations grappled with higher mortality rates due to cardiac and cerebral vascular diseases [4,5,6,7,8]. Our study, in line with prior research, highlights the need for a comprehensive investigation in the general community. Comparing our findings with the hospital’s report revealed significant disparities in the causes of mortality. Gastrointestinal issues were the leading cause, followed by cancer-related, respiratory, sepsis, cerebrovascular, cardiovascular, and urinary problems. Interestingly, diabetic patients showed a different pattern, with higher urinary tract-related mortality. The diabetic patients also exhibited lower cerebrovascular and cardiovascular mortality rates, possibly due to nationwide specialized centers. This observation is supported by the hospital’s annual reports, indicating a reduction in the incidence of these conditions in overall mortality causes. This underscores the urgent need for a study comparing mortality causes in people with and without diabetes in Mongolia’s general community. Until then, hospitals must remain vigilant, especially with diabetic patients, to promptly identify potential infections leading to sepsis. Recent studies have provided crucial insights into the primary cause of mortality in diabetes, emphasizing the pivotal role of infections and their complications [15,16]. Nevertheless, this factor exhibits significant variability among different countries. Some nations have experienced a decline in pneumonia cases, a trend attributed to regular immunization practices among patients [17,18]. Conversely, higher incidences of urinary tract infections and complications like DKA have been noted, which are potentially linked to the use of SGLT-2I [19,20]. In our study, a distinct pattern emerged among our patients. Despite lower immunization rates, pneumonia did not emerge as the primary concern. Moreover, we observed fewer cases related to urinary tract infections, even with the limited utilization of SGLT-2I. Instead, the most significant and common causes for concern were skin and soft tissue infections. While previous studies acknowledged the prevalence of skin and soft tissue infections [21], our findings highlighted a unique trend. Our cases were predominantly associated with skin-related issues such as cellulitis rather than complications such as osteomyelitis or postoperative wound infections. This trend could be attributed to poor hygiene practices prevalent in our population and inadequate glycemic control. Several studies have suggested that diabetics are more susceptible to skin and soft tissue infections due to alterations in both innate and acquired immune defenses which are often exacerbated by poor glycemic control [21,22]. Understanding these specific infection patterns is essential for tailoring effective preventive measures. Addressing hygiene practices and enhancing glycemic control could substantially improve the overall health outcomes for diabetic individuals in our community.

On a broader scale, our data imply that the distribution of the contributory causes of mortality generally aligns between participants with T1DM and T2DM, with minimal disparities across most causes. It is imperative, however, to approach these findings cautiously due to the relatively limited sample size for certain causes (e.g., cerebral vascular accidents and cardiomyopathy/heart blocks), which could impact the statistical reliability of our analysis. The relatively higher prevalence of chronic renal failure in T2DM patients may be attributed to the longer duration of diabetes in this group. Additionally, the prevalence of chronic liver failure in both T1DM and T2DM patients could be associated with Mongolia-specific conditions, possibly linked to a heightened incidence of viral liver infections in the country [23].

Expanding our investigation, we explored potential correlations involving variables such as age, gender, diabetes duration, smoking, and BMI status. Unfortunately, this exploration did not yield definitive outcomes. Nevertheless, a descriptive analysis hinted at the possibility that females might experience diabetes-related mortality at an older age despite having a shorter disease duration. This aligns with existing research highlighting gender disparities which often render females more susceptible to increased severity and mortality [24,25]. While sepsis emerged as the prevailing cause of mortality in our findings, we delved further into identifying specific risk groups. Interestingly, our analysis revealed that individuals with lower BMIs, smoking, and suboptimal glycemic control might face an elevated risk. This observation is consistent with prior research that has also emphasized the connection between a lower BMI and heightened mortality risk, a trend substantiated by previous studies [26,27]. A recent meta-analysis involving the general population revealed that smokers have a higher risk of mortality among septic patients [28]. Notably, poor glycemic control emerged as a primary risk factor for sepsis, a finding that resonates with various studies [29,30]. Our study further contributes to this evolving understanding.

We initially anticipated observing elevated mortality rates in Mongolia attributed to hypoglycemia and DKA. However, intriguingly, our findings did not identify any instances of mortality linked to hypoglycemia. Moreover, the mortality associated with DKA was lower than expected, indicating potential enhancements in the Mongolian healthcare system. This trend aligns with global observations from other studies, which demonstrate a decreasing trend in mortality related to acute diabetic complications on a worldwide scale [31,32]

It is important to acknowledge the limitations of our study when interpreting the results. Firstly, the sample size was constrained due to the reliance on data from a single tertiary care hospital in Mongolia, potentially affecting the generalizability of our findings. Nevertheless, it is worth highlighting that this hospital is a significant institution catering to patients from both rural and urban areas, functioning as a nearly fourth-level institution in the country’s healthcare hierarchy. Additionally, the retrospective study design, which is reliant on pre-existing data, introduces inherent biases and potential gaps in information. Furthermore, the lack of multicenter data could restrict our ability to capture variations in mortality causes across diverse healthcare centers or regions. Notably, the recent establishment of a hospital unit focused on cerebrovascular accidents might have led to the omission of patients redirected there, potentially overlooking cases that were not within the scope of our study. Despite these limitations, our study boasts notable strengths. It leveraged a comprehensive dataset spanning four years, facilitating a thorough analysis of mortality causes in diabetic patients. Moreover, its relevance to developing countries like Mongolia offers insights that can steer targeted enhancements in healthcare strategies for the management of diabetes in analogous contexts. Through the implementation of a hospital registration system, our findings can furnish valuable insights which are applicable to developing nations, catalyzing improvements in diabetes management and healthcare systems, thereby curbing mortality rates on a broader scale.

## 5. Conclusions

In Mongolia, this analysis of hospital-based data underscores sepsis as the leading cause of mortality among diabetic patients admitted to tertiary hospitals, irrespective of age, gender, or the type of diabetes. Notably, a potential link between a lower BMI, inadequate glycemic control, smoking, and the risk of sepsis emerged. To reinforce these findings, there is a pressing need for comprehensive studies across all hospitals in Mongolia, emphasizing the necessity of establishing a nationwide diabetes registration system.

## Figures and Tables

**Figure 1 jcm-12-06687-f001:**
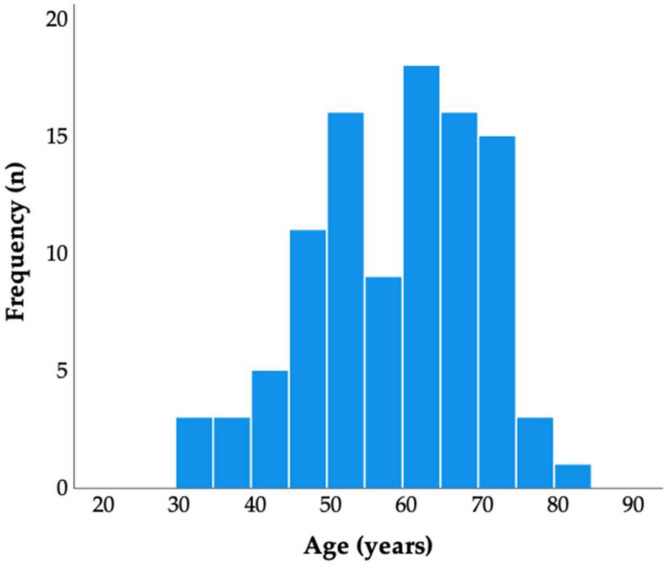
Histogram of the age of diabetes patients.

**Figure 2 jcm-12-06687-f002:**
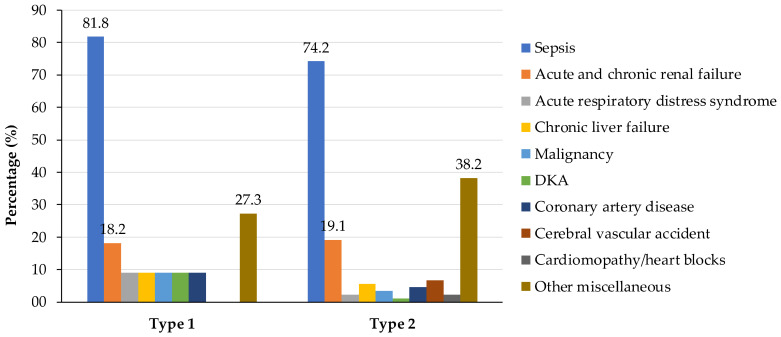
Causes of mortality in diabetic patients by type of diabetes.

**Figure 3 jcm-12-06687-f003:**
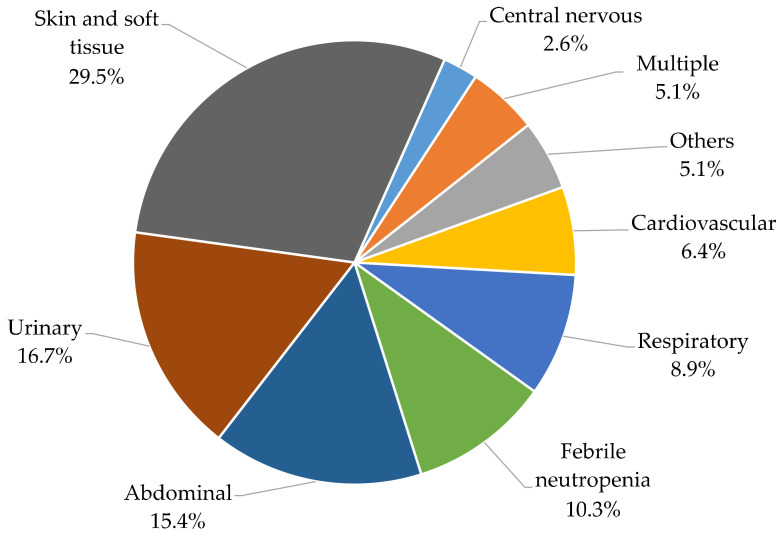
Remote causes of sepsis.

**Table 1 jcm-12-06687-t001:** Single and contributory causes of mortality in diabetic patients.

	Single Cause (n = 100)	Contributory Cause, % (n)			
		Total (n = 100)	Men (n = 53)	Women (n = 47)	*p*-Value
Sepsis	65.0 (65)	75.0 (75)	77.4 (41)	72.3 (34)	0.364
Acute and chronic renal failure	7.0 (7)	19.0 (19)	15.1 (8)	23.4 (11)	0.211
Acute respiratory distress syndrome	1.0 (1)	3.0 (3)	3.8 (2)	2.1 (1)	0.278
Chronic liver disease	1.0 (1)	6.0 (6)	3.8 (2)	8.5 (4)	0.283
Malignancy	2.0 (2)	4.0 (4)	5.7 (3)	2.1 (1)	0.355
Cerebral vascular accident	3.0 (3)	6.0 (6)	3.8 (2)	8.5 (4)	0.283
Cardiomyopathy/heart blocks	2.0 (2)	2.0 (2)	1.9 (1)	2.1 (1)	0.722
Coronary artery disease	1.0 (1)	5.0 (5)	5.7 (3)	4.3 (2)	0.557
Diabetic ketoacidosis	2.0 (2)	3.0 (3)	3.8 (2)	2.1 (1)	0.278
Other miscellaneous	6.0 (6)	37.0 (37)	39.6 (21)	34.0 (16)	0.376
Multiple causes	10.0 (10)	-	-	-	-

Data are presented as percentages, % (numbers).

**Table 2 jcm-12-06687-t002:** The general characteristics of the study population.

Findings	Total (n = 100)	Male (n = 53)	Female (n = 47)	*p*-Value
Age (year)	58.5 ± 11.3	56.1 ± 11.9	61.3 ± 10.1	**0.022**
Civilization: rural (%, n)	63.0 (63)	54.7 (29)	72.3 (34)	0.053
Education: lower (%, n)	10.0 (10)	11.3 (6)	8.5 (4)	0.449
Diabetes duration (year)	9.6 (0.81)	10.0 (1.12)	7.5 (0.92)	0.053
Type of diabetes: T1DM (%, n)	11.0 (11)	9.4 (5)	12.8 (6)	0.415
Body mass index (kg/m^2^)	27.1 ± 4.3	25.9 ± 4.2	28.4 ± 4.9	**0.009**
Glycated hemoglobin (%)	9.7 ± 3.6	10.1 ± 3.9	9.5 ± 3.3	0.459
Hospital beds (days)	12.5 (1.01)	14.2 (3.1)	10.6 (1.33)	0.308
Smoking status: smoker (%, n)	51.0 (51)	67.9 (36)	31.9 (15)	**<0.001**
Hypertension: Hypertensive (%, n)	61.0 (61)	54.7 (29)	68.1 (32)	0.122
Diabetes treatment:				
OGD	36.0 (36)	39.6 (21)	31.9 (15)	
OGD + insulin	49.0 (49)	43.4 (23)	55.3 (26)	0.489
Insulin	15.0 (15)	17.0 (9)	12.8 (6)	
Type of OGD				
Biguanide	61.0 (61)	60.4 (32)	61.7 (29)	0.528
Sulfonylurea	45.0 (45)	41.5 (22)	48.9 (23)	0.293
Incretin-based	20.0 (20)	18.9 (20)	21.3 (10)	0.479
SGLT-2I	8.0 (8)	7.5 (4)	8.5 (4)	0.573

Data are presented as means ± SDs or means (standard errors of the mean) and numbers (percentages, %). Bold values denote statistical significance at the *p* < 0.05 level. Note: SD, standard deviation; OGD, oral glycemic drug.

**Table 3 jcm-12-06687-t003:** Clinical characteristics in sepsis vs. non-sepsis patients.

Findings	Sepsis (+) (n = 75)	Sepsis (−) (n = 25)	*p*-Value
Age (year)	57.5 ± 11.2	61.7 ± 11.2	0.955
Gender: Male (%, n)	54.7 (41)	48.0 (12)	0.364
Civilization: rural (%, n)	61.3 (46)	68.0 (17)	0.352
Education: lower (%, n)	14.6 (11)	24.0 (6)	0.449
Diabetes duration (year)	8.2 (0.83)	10.0 (1.73)	0.726
Type of diabetes: T1DM (%, n)	12.0 (9)	8.0 (2)	0.447
Body mass index (kg/m^2^)	26.7 ± 4.1	28.5 ± 6.2	**0.014**
Glycated hemoglobin (%)	9.8 ± 3.1	9.6 ± 5.1	**0.008**
Hospital beds (days)	10.6 (1.61)	18.2 (5.01)	**0.004**
Smoking status: Smokers (%, n)	57.3 (8)	32.0 (8)	**0.024**
Hypertension: Hypertensive (%, n)	62.7 (47)	56.0 (14)	0.359
Diabetes treatment:			
OGD	36.0 (27)	36.0 (9)	
OGD + insulin	46.7 (35)	56.0 (14)	0.491
Insulin	17.3 (13)	8.0 (2)	
Type of OGD			
Biguanide	61.3 (46)	60.0 (15)	0.544
Sulfonylurea	41.3 (31)	56.0 (14)	0.148
Incretin-based	18.7 (14)	24.0 (6)	0.376
SGLT-2I	8.0 (6)	8.0 (2)	0.643

Data are presented as means ± SDs or means (standard errors of the mean) and numbers (percentages, %). Bold values denote statistical significance at the *p* < 0.05 level. Note: SD, standard deviation; OGD, oral glycemic drugs.

## Data Availability

The data used to support the findings of this study are available from the corresponding author upon request.

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
