# Peer review of "Diabetes-Related Mortality in a Developing Country: An Exploration of Tertiary Hospital Data"

_jcm, 2023, doi:10.3390/jcm12206687_

Round 1
Reviewer 1 Report
Dear authors,
Thank you for this interesting study that talk about diabetes-related mortality in Mongolia.
I have only few concerns about the form: DKA must be defined earlier in the text, as you use this acronym line 175 but explain it line 251.
Otherwise, concerning the study itself, sepsis seems to be a major cause of death in your patients with diabetes. This is actually also what can be noticed in other countries, and sepsis is always a matter of concern for patients with diabetes.
I think nevertheless you could have precised, if possible concerning the retrospective aspect or your study, the different kinds of infections that led to a sepsis (pulmonary infections, skin infections, bone and joint infections etc). A cohort study published in 2018 (Carey et al, diabetes care 2018;41:513-521) had compared the risk of infections between diabetic and non diabetic people, and patients with diabetes had a higher risk of bone and joints infections, but also endocarditis, meningitis, pneumonia, cellulitis etc. Depending on the leading cause of infections, some preventive measures could be reinforced (like vaccination if pulmonary infections due to pneumococcus are frequent etc).
The diabetes Fremantle study also found a higher risk of hospitalization for sepsis in patients with diabetes (Hamilton et al, PloS one 2013).
So these informations about sepsis in patients with diabetes are already well-known, if you could analyse more precisely the leading causes of infections, I think the study and article would be more powerfull and could led to adapted preventive measures that could really be interesting for your population.
In the Table 1, which shows the general characteristics of the population, I think that some other factors (again if possible) could be precised, such as the diabetes' treatments, the smoking status or maybe the presence or not of hypertension.
Indeed, if patients are all (or almost) treated with GLP1 agonists since a few years, it could be taken into account in the relative "low" level of coronary artery disease events and cerebro-vascular disease events you find in the study. On the other hand, if patients are frequently treated with SGLT2 inhibitors, it could also be taken into account in the analyses about sepsis, especially if sepsis is often due to genito-urinary infections.
Active smoking and hypertension are also other cardiovascular risk factors that could influence the occurrence of coronary artery disease or cerebrovascular diseases.
All these factors could be taken into account in your analyses.
Author Response
We express our sincere gratitude for your invaluable comments, which have undeniably contributed to the enhancement of our manuscript.

Reviewer 2 Report
This is a well presented descriptive study by Sankhuu et al. The authors describe causes of death within diabetes in a tertiary hospital in Mongolia.
One main comment/question remains for me
- What type of patients are generally referred to this tertiary hospital. Is there certain expertise in this hospital that may lead to overrepresentation of diseases (and thereby causes of death) in these patients?
- How do the causes of death within persons with diabetes compare to causes of death in the hospital in general? Or to the causes of death in Mongolia in general?
This will shed light on whether persons with type 2 diabetes actually relatively die more/less often from certain causes which is the goal the authors had in mind with the paper.
Author Response

(The authors gave the same response as above.)
